

# Subharmonic resonant excitation of edge waves by breaking surface waves

Nizar Abcha[1], Tonglei Zhang[1], Alexander Ezersky[1], Efim Pelinovsky[2-4] and Ira Didenkulova[2,5]

[1]Morphodynamique Continentale et Côtière UMR6143, CNRS, Normandie Univ, UNICAEN, 14000 Caen, France
5 [2]Nizhny Novgorod State Technical University n.a. R.E. Alekseev, 24 Minin Str., Nizhny Novgorod 603950, Russia
[3]Institute of Applied Physics, 46 Uljanov Str., Nizhny Novgorod 603950, Russia
[4]National Research University – Higher School of Economics, Nizhny Novgorod 603950, Russia
[5]Marine Systems Institute at Tallinn University of Technology, Akadeemia tee 15A, 12618 Tallinn, Estonia

*Correspondence to*: Nizar Abcha (nizar.abcha@unicaen.fr)

10 **Abstract.** Parametric excitation of edge waves with a frequency two times less than the frequency of surface waves propagating perpendicular to the inclined bottom is investigated in laboratory experiments. The domain of instability on the plane of surface wave parameters (amplitude-frequency) is found. The subcritical instability is observed in the system of parametrically excited edge waves. It is shown that breaking of surface waves initiates turbulent effects and can suppress the parametric generation of edge waves.

## 1 Introduction

Parametric excitation of waves with a half of external frequency has a long history. First papers on this subject were published by M. Faraday who described excitation of capillary ripples with a frequency $\Omega/2$ in a thin horizontal layer of viscous fluid placed on a horizontal plate oscillating vertically with a frequency $\Omega$ (Faraday 1831). After Faraday, such parametric excitation of waves have been observed in hydrodynamics (Dauady 1990; Cerda and Tirapegui 1998), plasma physics (Okutani et al. 20 1967; Kato et al. 1965), chemically active media (Fermandez-Garcia 2008) and other systems. Such parametric excitation also occurs in the ocean. Surface waves approaching the shore from the open sea with a frequency $\Omega$ can excite the so-called edge waves with a frequency $\Omega/2$. Edge waves propagate along the coastline with their amplitudes decreasing in offshore direction (Ursell 1952; Grimshaw 1974; Guza and Davis 1974; Evans and McIver, 1984; Johnson 2005, 2007). Interest in parametrically excited edge waves is caused by their ability to significantly affect morphological characteristics of sea coasts. Edge waves 25 may contain enough energy to be responsible for the erosion of the shore. They may also focus forming a freak wave (Pelinovsky et al. 2010). Sometimes edge waves are also associated with the generation of beach cusps (Guza and Imman 1975; Komar 1998; Masselink 1999; Dodd et al. 2008; Coco and Murray 2007).

Analytical solutions for edge waves excited by nonbreaking surface waves are obtained in (Akylas 1983; Minzoni and Whitham 1977; Yeh 1985; Yang 1995, Blondeaux and Vittori 1995; Galletta and Vittori 2004; Dubinina et al. 2004). The 30 correlation between characteristics of edge waves and spectra of surface waves approaching the shore are studied *in situ*



(Huntley and Bowen 1978). This kind of studies is complicated for analysis and result interpretation due to the irregularity of the coastline and complex spectra of the approaching surface waves.

Laboratory experiments on parametric excitation of edge waves are described in (Buchan and Pritchard 1995). The main advantage of such experiments is the freedom to define the bottom geometry and spectrum of the approaching surface waves.

However, none of the studies mentioned above considered wave breaking, whereas in natural conditions surface waves often break while propagating towards the coastline. Thus, the influence of wave breaking on a parametric instability still remains a question. In the present paper, we concentrate on influence of wave breaking on characteristics of parametrically excited edge waves.

The paper is organized as follows. In Section 1 we focus on the theoretical description of the problem providing the nonlinear

equation for parametric excitation of edge waves. Section 2 is devoted to the experimental set-up, while Section 3 presents the results of measurements. In section 4, we discuss the experimental data in view of their theoretical interpretation. The main results are summarized in Conclusion.

## 2 Theoretical model

Let us start from the non-breaking scenario, when long waves propagate over some changing bottom geometry $h = h(x)$. In

this case they can be described by 2D nonlinear shallow water equations:

$$\frac{\partial u}{\partial t} + u\frac{\partial u}{\partial x} + v\frac{\partial u}{\partial y} + g\frac{\partial \eta}{\partial x} = 0, \tag{1}$$

$$\frac{\partial u}{\partial t} + u\frac{\partial v}{\partial x} + v\frac{\partial v}{\partial y} + g\frac{\partial \eta}{\partial y} = 0, \tag{2}$$

$$\frac{\partial \eta}{\partial t} + \frac{\partial}{\partial x}\left(u(h+\eta)\right) + \frac{\partial}{\partial y}\left(v(h+\eta)\right) = 0, \tag{3}$$

where $(u, v)$ are the two components of the depth-averaged horizontal velocity, $\eta = \eta(x, y, t)$ is the free surface displacement,

and $g$ is the gravity acceleration. In a linear approximation the system (1) - (3) can be transformed into 2D wave equation:

$$\frac{\partial^2 \eta}{\partial t^2} - g\left(\frac{\partial}{\partial x}\left[h(x)\frac{\partial \eta}{\partial x}\right] + h(x)\frac{\partial^2 \eta}{\partial y^2}\right) = 0. \tag{4}$$

Note, that equation (4) describes both surface waves propagating perpendicular to the shore and generated edge waves. For the edge waves we assume that they propagate along the shore, and consider a linear change of the bottom slope $h(x) = \beta x = \tan\alpha \, x$. In this case an elementary solution of equation (4) has the following form:

$$\eta = b\cos(\Omega_n t - ky)\cdot e^{-kx}L_n(x), \qquad \Omega_n = \sqrt{(2n+1)\beta gk}, \qquad n = 0,1,2,\ldots \tag{5}$$

where $L_n$ are the Laguerre polynomials, $b$ is a wave amplitude, $k$ is a wave number along the propagation direction, $\Omega$ is a wave frequency, and $n$ is the number of the mode.



Using two edge waves propagating in opposite directions, it is also possible to compose a solution corresponding to a standing edge wave, with boundary conditions $v(x, y, t) = 0$ for $y = \pm L/2$, where $L$ is a channel width:

$$\eta = b\cos(\Omega_{n,m}t)\sin(k_m y)L_n(x)\,,\quad k_m = \pi(1+2m)/L\,,\quad \Omega_{n,m} = \sqrt{(2n+1)\beta g k_m}\,,\quad m = 0,1,2,..,\tag{6}$$

For surface waves propagating perpendicular to the shore equation (4) transforms into a 1D wave equation:

$$\frac{\partial^2\eta}{\partial t^2} - g\frac{\partial}{\partial x}\left(h(x)\frac{\partial\eta}{\partial x}\right) = 0\,,\tag{7}$$

and has a solution in the form

$$\eta(x,t) = a_0 J_0\left(\sqrt{\frac{4\omega^3 x}{g\beta}}\right)\cos(\omega t)\,,\tag{8}$$

where $J_0$ is the Bessel function of the first kind, $\omega$ is a frequency and $a_0$ is an amplitude of the generated surface wave.

In a linear approximation waves (6) and (8) are independent. If nonlinear effects are taken into consideration [equations (1) - (3)], coupling between the two types of waves takes place. In the first approximation of nonlinearity, surface waves described by (8) can generate edge waves described by (6) if $\Omega \approx \omega/2$. It is the so-called parametric subharmonic resonance. In this case, we can write down the equation for slowly varying wave amplitude $b$ of the excited edge waves with frequency $\Omega$ (Rabinovich et al 2000):

$$\frac{\partial b}{\partial t} = -\gamma b + \mu b^* + i\Delta b + (i\sigma - \rho)b|b|^2\,.\tag{9}$$

Here $\gamma$ represents an exponential decay of edge waves due to the viscous dissipation, $\Delta = \Omega - \omega/2$ is a detuning between frequencies of edge waves and the external parametric forcing, $\sigma$ is a nonlinear frequency shift, $\rho$ is a nonlinear damping coefficient. This equation was initially obtained for Faraday ripples excited by a homogeneous oscillating field. For edge waves excited by surface waves propagating perpendicular to the shore, an expression for a coefficient $\mu$ has been obtained in (Akylas 1983; Minzoni and Whitham 1977; Yang 1995):

$$\mu = a_0\frac{\omega^3}{4g\beta^2}S(\beta)\,.\tag{10}$$

Here $S$ is a coefficient depending on a bottom slope $\alpha$. For small slopes $\alpha$, $S \approx 6.7 \; 10^{-2}$. The nonlinear frequency shift $\sigma$ has been calculated in (Minzoni and Whitham 1977). The nonlinear damping coefficient $\rho$ has been discussed in (Yang 1995).

## 3 Experimental set-up

Experiments have been performed in the wave flume of the Laboratory of Continental Coastal Morphodynamics of the Caen University, France. This flume has length of 18 meters and width of 0.5 m. The flume is equipped with a wave-maker controlled by the computer. For construction of an inclined bottom slope a PVC plate of 0.01 m of thickness has been used. The plate has



been placed at an angle $\alpha$ to the horizontal bottom so that $\tan \alpha = \beta = 0.20$; the water depth in the flume, $h$ has been kept at 0.25 m (see Fig. 1). As one can see from Fig. 1, in this geometric configuration only a small part of the flume could be used for experiments. Three resistance probes P1, P2, P3 (see Fig. 1) have been used to measure a water surface displacement. The first of them, the immobile probe P1 have been placed at a distance of 1 cm from the wave maker, while probes P2 and

P3 have been glued to the inclined plate. The latter two probes placed along the bottom slope allow us to measure wave run-up and run-down. In addition, the run-up height can be identified by image processing from the high-speed camera operating with a frame rate of 100 Hz (see Fig. 1). The wave maker oscillating with a given frequency and amplitude allows to excite the targeted mode described by equation (8). The wave maker can work in two regimes. The first regime controls the amplitude of the wave maker displacement, while the second one controls the amplitude of the force applied to the wave maker. In both

regimes it is not possible to control the free surface displacement. Therefore, to study the surface wave characteristics, simultaneous measurements of a free surface displacement near the wave maker and the shoreline have been carried out. For velocity fields (all three components of the flow velocity), the Acoustic Doppler Velocimeter (ADV) has been used. The quality of the signal registered by ADV strongly depends on the concentration of particles in the liquid. Therefore, in order to get a better signal, some small particles with diameter of 10 μm have been added into the water.

For visualization of a free surface displacement in the breaking zone by the high-speed camera, the water has also been seeded with sand particles of 10 μm. Using a vertical light sheet (photodiode 532 nm with spherical lens) it has been possible to visualize the cross-section of the water in the $x$-$z$ plane. The size of the visualization domain stands for 40 cm × 30 cm.

## 4 Data processing and results

The subharmonic instability described above has been investigated in the flume for different values of $(a_L, f)$, where $a_L$ is the

amplitude of surface waves in the vicinity of the wave maker, $a_L \approx a_0$, and $f$ is the frequency of the wave maker. In order to understand whether instability really occurred, we analyzed the signals from probes P2 and P3. Before each experiment we had been waiting for 5 – 10 min to let all the perturbation in the flume decay, and let wave maker work in calm water conditions. An example of signals from P2 and P3 is shown on Fig. 2a, whereas a more detailed zoom of the time series for intervals 50 s < $t$ < 95 s and 85 s < $t$ < 90 s is given in Fig. 2b and Fig. 2c respectively.


It can be seen that in the beginning of the record the waves have the same frequency and phase as the wave maker (Fig. 2b). However, after instability arises (Fig. 2c) the amplitude of generated edge wave increases and the period doubles compare to the period of surface waves. The phase shift between the signals recorded by probes P2 and P3 is approximately π. These two criteria (period doubling and a phase shift equal to π) are used to identify parametric instability. To confirm an appearance of

edge waves as a result of subharmonic instability, we perform analysis of water level oscillations. It is found that subharmonic oscillations represent the mode: maxima of horizontal displacement (antinodes) occur near the lateral walls of the flume, while its zeroes (nodes) are observed in the middle of the flume. This mode is a superposition of two edge waves propagating in





opposite directions. A spatial period of these edge waves is twice larger than the width of the flume. Snapshots of water surface through the time interval equal to half of the edge wave period are shown in Fig. 3.

Subharmonic instability starts with an exponential growth of an infinitely small perturbation. To describe the instability in the system, partition of a $(a_L, f)$ plane into different stability regions has been performed. Results of this analysis are demonstrated

in Fig. 4.

Instability occurs if the frequency of surface waves is close to a double frequency of edge waves. Curve 1 represents a border of supercritical instability regime which occurs for points $(a_L, f)$ above this curve. If amplitude of surface waves decreases from a finite value above Curve 1, generation of edge waves is observed in a small region (3) between Curves 1 and 2 (see, triangles in Fig. 4). When we start from the regime without edge wave generation (points below Curve 2) and increase the

amplitude of surface waves, instability will occur above Curve 1. This type of instability is called subcritical instability.

The partition of a plane $(a_L, f)$ into regions with different regimes shown in Fig. 4 corresponds to two qualitatively different conditions of wave excitation schematically shown by boxes (I) and (II). In the Region I surface waves excited by the wave maker and propagating to the shore undergo a plunging wave breaking. In the Region II waves do not break. Image processing of high-speed camera data shows that such excitation occurs only when the wave breaking parameter $Br > 0.9$. Under the wave

breaking parameter we mean $Br = U_{\max}^2 / gR$, where $U_{\max}$ is the maximal flow velocity, and $R$ is the maximal wave run-up height on the shore (Didenkulova 2009).

It is found that while surface wave breaking leads to appearance of hydrodynamic turbulence, turbulence itself leads to decrease in the amplitude of excited edge waves and suppression of subharmonic generation for large amplitude surface waves.

Dependences of the increment of edge wave instability and intensity of turbulent velocity fluctuation on the amplitude of

surface waves $a_L$ are shown in Fig. 5a and 5b. The dependence of the exponential index the same $\gamma$ on the amplitude of surface waves $a_L$ is found by processing corresponding time series similar to those shown in Fig. 2a. For this we selected time intervals where the edge wave amplitude grows and calculated $\gamma$ by exponential approximation of the time dependent amplitude.

Parameters of turbulence are measured by ADV in the middle of the experimental flume, 0.04 m below the free surface (0.14 m

from the bottom), at a distance $x = 0.9$ m from the shoreline. In this point it is possible to neglect the turbulence caused by the near-bottom oscillating boundary layer and detect the wave breaking turbulence.

Here we should specify some difficulties related to the characteristic features of ADV signals. The recorded ADV signals contained the so-called spikes, which have been filtered using the MATLAB algorithm by (Nikora and Goring 1998; Goring and Nikora 2002). Another problem was due to the complex structure of the velocity field in the breaking zone, which

represents a mixture of turbulence and velocities caused by both surface and edge waves. In this case, impact of surface and edge wave components have been removed by filtering harmonics with frequencies $f/2$, $f$, $3f/2$, $2f$, $5f/2$ and $3f$. The measurements show that the intensity of turbulence grows sufficiently if the amplitude of surface wave $a_L$ is larger than 0.8 cm, see Fig. 5b.





## 5 Discussion

The region of parameters corresponding to the parametric excitation of edge waves has been found experimentally. Now, using the theoretical formula (10), we can estimate the threshold of parametric excitation of edge wave. For this we need to find the eigenfrequencies of edge waves in the flume $\Omega_n$. The frequency of the zero edge wave mode $\Omega_0$ has minimum dissipation:

$$\Omega_0 = \sqrt{\beta g \frac{\pi}{L}} = 3.41 \text{ rad/s}, \qquad f_0 \approx 0.54 \text{ Hz}. \tag{11}$$

To estimate the dissipation rate of edge waves, we study the time evolution of the edge wave amplitude after stopping the parametric excitation. Edge waves decay exponentially and in this way we measure the decay rate $\gamma$, which is estimated as $\gamma = 0.1$ s$^{-1}$. For the resonance condition $\Delta = 0$, parametric instability occurs when wave amplitude exceeds the critical wave amplitude $a_0$:

$$a_0 = \gamma \frac{4g\beta^2}{\omega^3 S(\beta)} \approx 0.76 \text{ cm}. \tag{12}$$

The theoretical value of the parametric instability threshold is calculated using the free surface displacement. To compare experimental and theoretical values of the threshold, we need to measure the surface wave amplitude at $x = 0$. As it has been noted in several studies, (see, for example, Denissenko et al. 2011), this value can be measured indirectly. We find it using flow visualization by the laser sheet, which had been carried out in the middle of the flume at a time preceding the development of the edge wave parametric instability (see Fig. 6).

Note, while the parametric instability threshold was determined, there was no surface wave breaking, which corresponded to the Region II in Fig. 4.

Fig. 6 has been shot before development of the parametric instability, when amplitudes of edge waves were zero. To estimate the surface wave amplitude, the measured crest-to-trough wave height (Fig. 6) has been divided by two. Comparison of the experimental and theoretical values of the instability threshold is shown on Fig. 7. One can see from Fig. 7 that theoretical values are larger than experimental ones by approximately 30%.

Note, even when the surface wave breaking takes place, parametric excitation of edge waves still occurs. However, parametric excitation is suppressed for large amplitudes of surface waves. The reason for this could be the following. The wave breaking results in the irregularity of the surface wave field: amplitudes and phases of the waves vary chaotically. Evidently, wave breaking also leads to the appearance of small scale turbulence in the nearshore zone. Below we discuss the impact of these two physical mechanisms on the suppression of parametric instability.

The parametric wave excitation by the irregular oscillating field has been studied in (Ezersky and Matusov 1994; Nikora et al. 2005). It was shown that chaotic amplitudes and phases of the external wave field lead to increase in the threshold of parametric excitation and decrease in the amplitude of parametrically excited oscillations.



Let us check whether these results can explain the decrease in the edge wave amplitude in presence of the wave breaking. For this we calculate amplitudes and phases of surface waves. After narrow band filtering, generated by the wave maker surface waves may be described as $\eta_m \cos(\omega t + \Phi)$, where $\eta_m$ is a slow varying amplitude, and $\Phi$ is a slow varying phase. To extract amplitude and phase of the signal, the Hilbert transformation is used:

$$\hat{\eta}(t) = \frac{1}{\pi} PV\left[\int_{-\infty}^{+\infty} \frac{\eta(t,\tau)}{t-\tau} d\tau\right] = \eta_m \sin(\omega t + \Phi), \tag{13}$$

where $PV$ denotes the principal value of the integral. It is also possible to determine wave amplitude and phase:

$$\eta(t) = \mathrm{Re}\{a(t)\exp(i\omega t)\}, \quad a(t) = |a|e^{i\Phi}, \tag{14}$$

where

$$|a| = \sqrt{\eta^2 + \hat{\eta}^2}, \quad \Phi = \arctan(\hat{\eta}/\eta) - \omega t. \tag{15}$$

Extracted amplitudes and phases for the time series measured in presence of the surface wave breaking are shown in Fig. 8. The time series itself is given in Fig. 8a, while amplitudes and phases are shown in Fig. 8b. The root mean square of phase and amplitude fluctuations for intensive wave breaking ($a = 1.4$ cm) is

$$\sqrt{\langle \Phi^2 \rangle} \approx 0.1, \quad \frac{\sqrt{\langle (a - \langle a \rangle)^2 \rangle}}{\langle a \rangle} \approx 0.1. \tag{16}$$

It is also possible to estimate the influence of chaotic phases and amplitudes on the parametric wave excitation. It has been revealed that chaotic phase decreases the effective amplitude of the external force (Petrelis et al. 2005). Suppose, that wave breaking leads to the Gaussian noise, then the corresponding decrease in the external forcing may be estimated as (Petrelis et al. 2005):

$$e^{-\langle \Phi^2 \rangle/2} \approx 0.995. \tag{17}$$

This small decrease in the effective external forcing cannot explain suppression of the parametric excitation during the wave breaking regime, therefore, the influence of the turbulence seems to be more important.

Wave breaking generates turbulence and the intensity of turbulent velocity fluctuations grows with the surface wave amplitude. On the other hand, turbulence leads to the appearance of turbulent shear stresses and eddy viscosity $\nu_{ed}$. We measure experimentally some components of the kinematic turbulent energy at the edge wave background (see Fig. 5b). According to our measurements the most important components of shear stresses are related to the longitudinal component of turbulent fluctuations $V_x$ (see Fig. 5b).

The eddy viscosity $\nu_{ed}$ is proportional to the turbulent energy. For the wave breaking case one can consider $\nu_{ed}$ to be proportional to $a^2$ (see Fig.5b). In this case the exponential decay decrement $\gamma$ in equation (9) has the following form: $\gamma = \gamma_0 + \gamma_1 a^2$, where $\gamma_0$ is the exponential decay the same of edge waves in the absence of wave breaking, and $\gamma_1$ is responsible for energy dissipation due to eddy viscosity.

Since the external forcing $\mu$ grows linearly with the surface wave amplitude and the dissipation is proportional to the amplitude squared, the parametric instability is suppressed for large surface wave amplitudes. This effect we observe in our experiment

under the surface wave breaking regime.

## 6 Conclusions

The parametric edge wave excitation is studied for different regimes of surface wave propagation. Found, that for parametrically excited edge waves there is a region of subcritical instability, which is manifested by the hysteresis: different regimes of edge wave excitation are observed in the case of decrease or increase of the surface wave amplitude. Note, that

subcritical instability was not observed in (Buchan and Pritchard 1995), though their experimental conditions were very close to those in our experiment.

The increase in the surface wave amplitude leads to the appearance of wave breaking. The wave breaking regime itself does not prevent parametric excitation of edge waves; only the developed wave breaking can suppress parametric excitation of edge waves. We compare two possible mechanisms of the parametric instability suppression: phase irregularity of the external

forcing and generation of the hydrodynamic turbulence. Found, that the most probable mechanism responsible for the increase of the parametric instability threshold and suppression of parametric excitation of edge waves is the hydrodynamic turbulence which appears as a result of wave breaking.

### Acknowledgments

This work is dedicated to Professor Alexander Ezersky who was the key author and the main driver of this study. Last summer he sadly passed away after a long lasting fight with the cancer leaving the manuscript unfinished. Until his last days he tried to dedicate his time to work, including the results presented here. Therefore, we took as a must to conclude his work in memory of a dear friend and colleague.

The present study was partially supported by RFBR grants (15-35-20563, 15-55-45053), a Russian President Grants MD-

6373.2016.5 and NS -6637.2016.5. ID and EP also thank University of Caen for their visitor program, which allowed this fruitful collaboration.

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





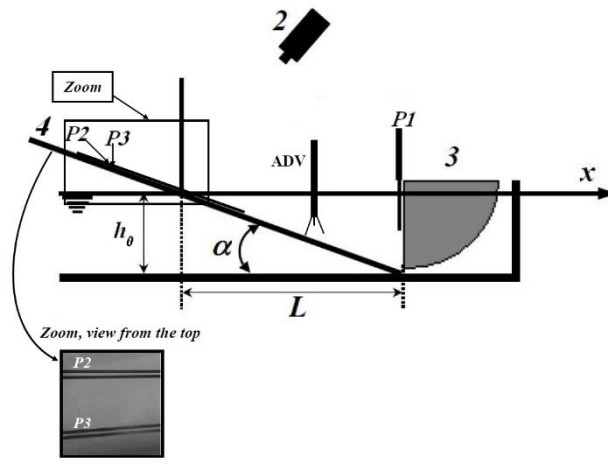

**Figure 1. The experimental set-up: resistance probes: vertical (P1) and horizontal (P2, P3), a high-speed video camera (2), a wave maker (3), an inclined bottom (4), and ADV.**





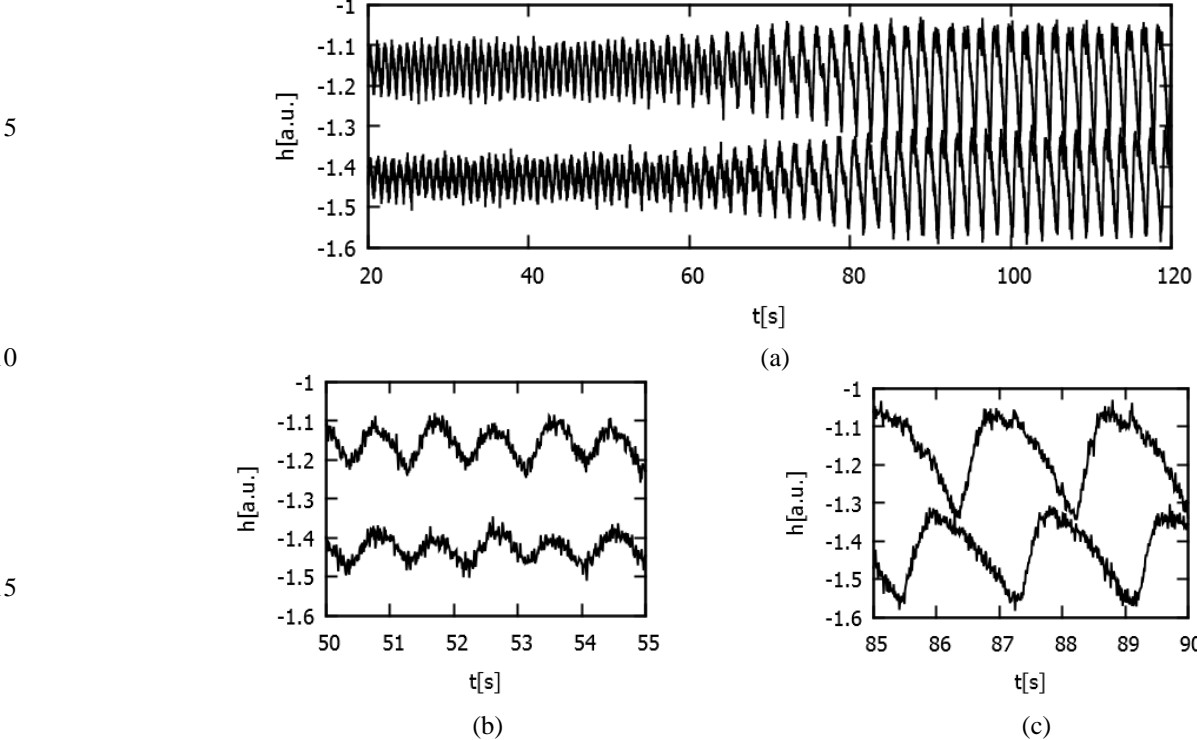

**Figure 2. Example of wave instability developing from a natural perturbation with** $f = 1.08$ **Hz,** $a_L = 0.66$ **cm: (a) the full time series recorded by probes P2 and P3; (b) zoom of the time series recorded during the time interval 50 s** $< t <$ **55 s, and (c) during the time interval 85 s** $< t <$ **90 s.**




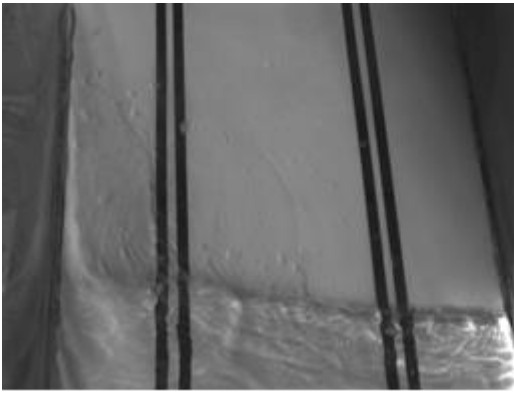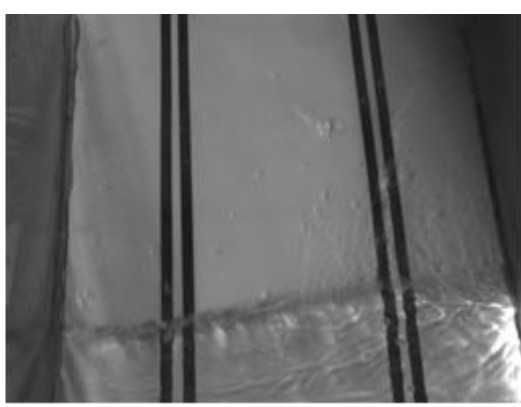

**Figure 3. Snapshots of water surface through the time interval equal to half of the edge wave period (approximately 1 s),** $f$ = 1.06 Hz, $a_L$ = 1.3 cm.









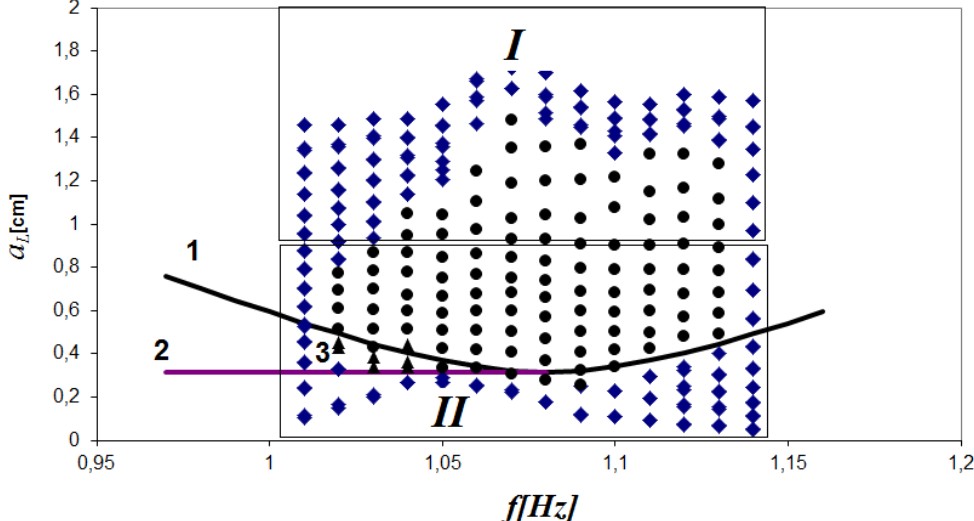

**Figure 4. Partition of a $(a_L, f)$ plane into different stability regions of the system; circles correspond to a parametric instability, diamonds correspond to stability regimes, and triangles are for the regime of subcritical instability.**





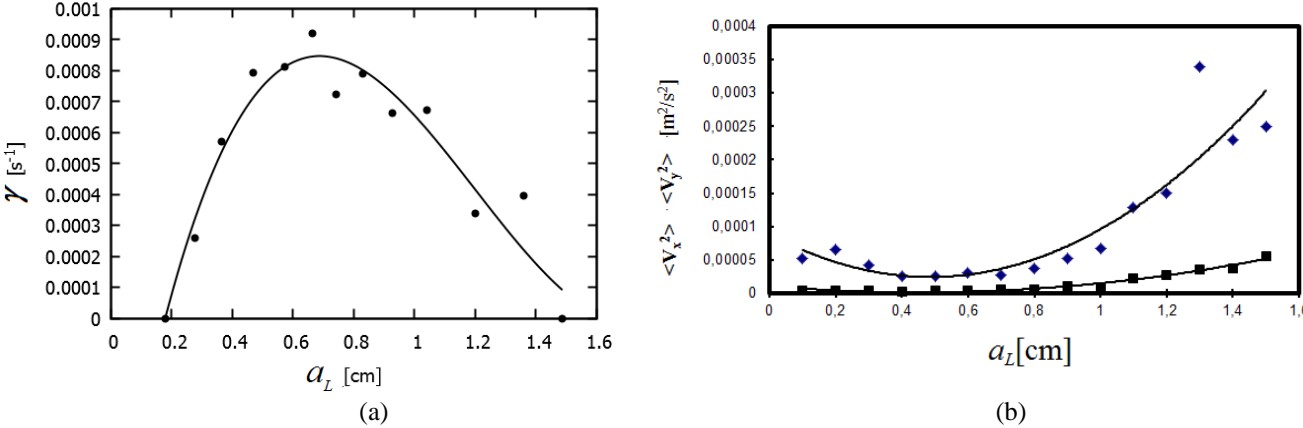

(a)  (b)

**Figure 5. (a) Dependence of the exponential index of parametric instability $\gamma$ on the surface wave amplitude $a_L$, shown by the black dots, and (b) dependence of the kinematic turbulent energy components on the surface wave amplitude $a_L$; $V_x$ is shown by blue diamonds, while $V_y$ is shown by black squares.**





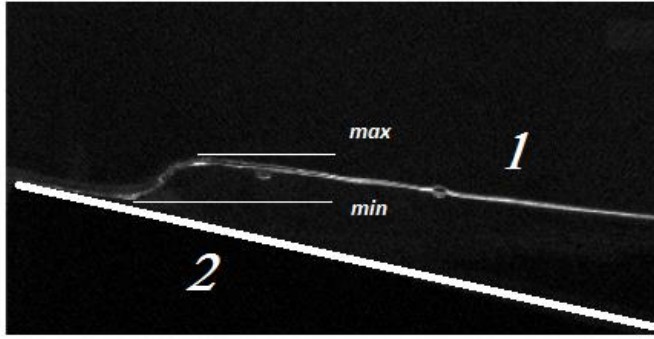

**Figure 6. Visualization of the free surface displacement: 1 is for the water surface, 2 is for the inclined bottom, *max* and *min* correspond to the maximum and minimum values of the free surface displacement.**









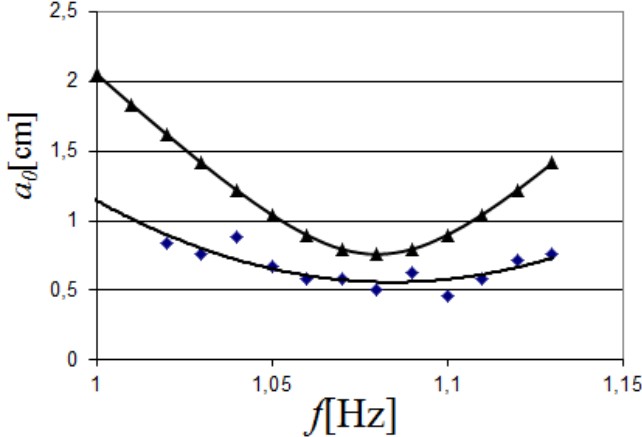

**Figure 7.** Comparison of experimental and theoretical values of the instability threshold: triangles correspond to the theoretical formula, diamonds represent experimental data.









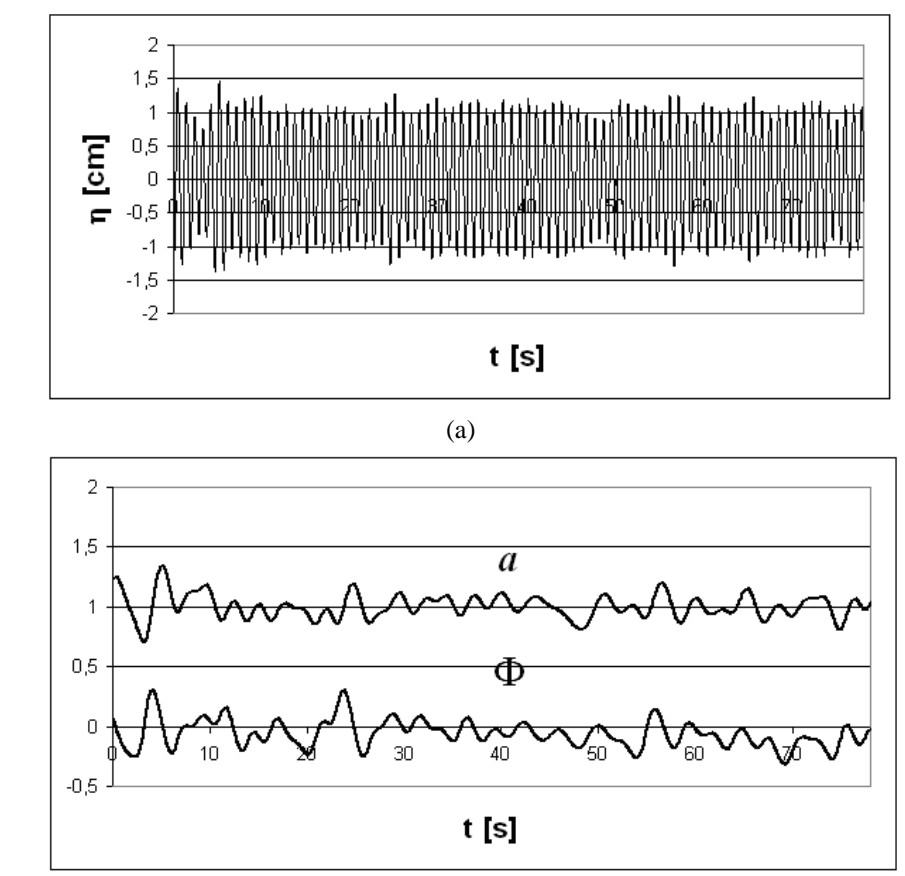

**Figure 8. (a) Time series measured by P1 with $a_L$=1 cm, $f$ =1.06 Hz; (b) non-dimensional wave amplitude and phase obtained by the Hilbert transformation.**