# Peer review of "Subharmonic resonant excitation of edge waves by breaking surface waves"

_Nonlinear Processes in Geophysics, 2016_

## Referee Comment (RC1) · Anonymous Referee #1 · 16 Nov 2016

The article touches a physically interesting phenomena of half-frequency edge waves. Results look interesting and useful in the context of shore wave dynamics.

A few points need to be expanded though.

1) It is worth presenting the raw data explicitly displaying the period doubling effect. ADV versus wave gauges? Difference between wave gauge reading?

2) The flume is narrow hence parameters of transverse oscillations are somewhat defined by its width. It is worth commenting on the choice of excitation frequency. It could happen that secondary waves may appear due to asymmetry of the wavemaker or some other parameters of the flume. Transverse waves do routinely appear in such flumes all the time and the mechanisms can vary.

[Figure]

3) How the influence of reflected waves is accounted for? Duration of the experiment is not that long so talking about rising/receding wave amplitude should be accompanied by discussion of the applicability of the assumption about the incident wave parameters. Well.. ideally, incident wave parameters should be measured by an array of wave gauges.

---

## Referee Comment (RC2) · Anonymous Referee #2 · 8 Dec 2016

This paper deals with the problem of experimental excitation of edge waves when incident water waves propagate on a sloping sea bottom. Emphasis is put on the influence of breaking wave of edge wave generation. It is an interesting paper. My comments are the following: (i) To have more confidence in claims' authors, it should be more useful to use frequency spectra of the surface elevation, namely to demonstrate quantitatively the period doubling and edge wave suppression. Generally turbulent underlying flows attenuate or cancel water waves. So, it is not surprising that breaking waves which generate turbulence may suppress edge wave excitation. (ii) In equation (9) specify b* (complex conjugate). (iii) In figure 1, plot axes z and y. (iv) What kind of wavemaker is used? (v) I assume that in figure 5 the solid lines fit the experimental data. (vi) The English must e improved.

,

---

## Author Comment (AC1) · 14 Dec 2016

We thank the reviewer for his thoughtful critiques of our manuscript. We have adopted all of his suggestions. Our point-by-point response to the comments and questions is given below.

(i) To have more confidence in claims' authors, it should be more useful to use frequency spectra of the surface elevation, namely to demonstrate quantitatively the period doubling and edge wave suppression. We have added a new Figure 3, where we show two frequency spectra. The first spectrum (Figure 3 a) is in absence of the breaking waves, where the first peak indicates the edge wave frequency and the second peak indicates the surface elevation frequency. The second frequency spectrum (Figure 3 b) is plotted in presence of breaking wave and indicates the suppression of

[Figure]

the peak for the edge wave frequency.

(ii) In equation (9) specify b* (complex conjugate). Added after Eq. (9): "b* is a complex conjugate"

(iii) In figure 1, plot axes z and y. Done.

(iv) What kind of wavemaker is used? Added on Page 3: "The flume is equipped with a piston type of wave-maker controlled by the computer", see also the caption to Figure 1.

(v) I assume that in figure 5 the solid lines fit the experimental data. The caption to Figure 5 (now 6) has been changed:

(vi) The English must be improved. The language has been corrected

Please also note the supplement to this comment:
http://www.nonlin-processes-geophys-discuss.net/npg-2016-63/npg-2016-63-AC1-supplement.pdf

[Figure]

**Figure 1. The experimental set-up: resistance probes: vertical (P1) and horizontal (P2, P3), a high-speed video camera (2), a wave maker of a piston type (3), an inclined bottom (4), and ADV.**

**Fig. 1.**

[Figure]

(a)

(b)

**Figure 3. Power spectrum frequency: (a) in absence of breaking waves: the first peak indicates the edge wave frequency, while the second peak indicates the surface elevation frequency; (b) in presence of breaking waves: the peak for the edge wave frequency is suppressed.**

**Fig. 2.**

[Figure]

**Figure 6. (a)** Dependence of the exponential index of parametric instability $\gamma$ on the surface wave amplitude $a_L$, shown by the black dots, and **(b)** dependence of the kinematic turbulent energy components on the surface wave amplitude $a_L$; $V_x$ is shown by blue diamonds, while $V_y$ is shown by black squares. Solid lines represent a fit to the experimental data.

**Fig. 3.**

---

## Author Comment (AC2) · 21 Dec 2016

We thank the reviewer for his thoughtful critiques of our manuscript. We have adopted all of his suggestions. Our point-by-point response to the comments and questions is given below.

1) It is worth presenting the raw data explicitly displaying the period doubling effect. ADV versus wave gauges? Difference between wave gauge reading?

We have added a new Figure 3, where we show two frequency spectra. The first spectrum (Figure 3 a) is the FFT of the signal shown in Figure 2a. This is a spectrum in absence of breaking waves, where the first peak indicates the edge wave frequency and the second peak indicates the surface elevation frequency. The second frequency spectrum (Figure 3 b) is plotted in presence of breaking wave and indicates the suppression of the peak for the edge wave frequency.

2) The flume is narrow hence parameters of transverse oscillations are somewhat defined by its width. It is worth commenting on the choice of excitation frequency. It could happen that secondary waves may appear due to asymmetry of the wavemaker or some other parameters of the flume. Transverse waves do routinely appear in such flumes all the time and the mechanisms can vary.

Our excitation frequency range was chosen following our published study about the physical simulation of resonant wave run-up on a beach (see: Physical simulation of resonant wave run-up on a beach, Nonlin. Processes Geophys., 20. (2013)). In this study we describe edge waves excited by the 3rd resonant mode of the system.

3) How the influence of reflected waves is accounted for? Duration of the experiment is not that long so talking about rising/receding wave amplitude should be accompanied by discussion of the applicability of the assumption about the incident wave parameters.

The actual duration of the experiment is 240s, but for better graphic representation of the signal we show just first 120s. For the same reason we do not show the signal P1 recorded next to the wavemaker. We observe oscillations as a sum of incident and reflected waves. However, we use the signal just after the transition time, where the total amplitude is twice larger than the incident wave amplitude.

4) Well..ideally, incident wave parameters should be measured by an array of wavegauges.

We cannot use probes (such as probe P1 on Figure 1) very close to the shoreline due to the low water depth. This is why we use probes P2 and P3. P2 and P3 are placed and glued to the inclined bottom slope that allows us to measure wave run-up and run-down.

Please also note the supplement to this comment:

http://www.nonlin-processes-geophys-discuss.net/npg-2016-63/npg-2016-63-AC2-supplement.pdf
* * *
[Figure]

**Fig. 1.**

---

## Referee Report (RR1)

The article contains theoretical and experimental investigation of generation of edge waves due to the parametric resonance in near-shore area.

The article is worth publishing after the Authors address points described below.

**Incident wave field.** As the wave maker did not have a system to damp waves reflected from the slope, the incident wave field ceased to be harmonic (and in fact predictable at all) a few seconds after the wave generation starts.
The whole flume is 18 metre length while only 1.25 m of it was in use.
Why have not authors installed the slope at the very end of the flume to bringing the travel time of the reflected wave (and hence the duration of the "clean" incident wave field) to above 20 seconds?
Can the authors comment on the influence of incident wave field irregularity on the results?

**Stability threshold.** Can the discrepancy in Fig. 8 be explained by formation of the edge waves near the panel of the wave maker? Can that effect be evaluated?

It looks as the use of the whole length of the wave flume mentioned above could have helped decoupling the edge waves formed at the slope and at those formed at the wave maker.

It would be good to compare the rate of edge wave growth (not only the stability range) with the results predicted by (10).

**Turbulence.** From the measurements, it would be good to calculate the Energy Dissipation Rate and compare it with other typical powers present in the system such as power pumped into various wave harmonics.

It would make a good illustration if the Authors present the power spectrum of the turbulent flow in log-log scale.

**Minor points**

P. 4, Line 18: probably, a cylindrical lens not spherical.

Fig. 5: Do markers easier to distinguish by making, say, diamonds empty and circles filled.

Fig. 6, caption: Turbulent Kinetic Energy not kinematic, right?

P. 6, line 10: "The frequency of the zero edge wave mode Omega_0 has a minimal dissipation"
The reviewer does not understand relation of that to the formula (11). Why dissipation?

---

## Author Response (AR2)

**Subharmonic resonant excitation of edge waves by breaking surface waves**

by Nizar Abcha, Tonglei Zhang, Alexander Ezersky, Efim Pelinovsky and Ira Didenkulova

We thank the reviewer for his thoughtful critiques of our manuscript. We have adopted all of his suggestions. Our point-by-point response to the comments and questions is given below.

*Detailed response to reviewer 2*

**Incident wave field.** As the wave maker did not have a system to damp waves reflected from the slope, the incident wave field ceased to be harmonic (and in fact predictable at all) a few seconds after the wave generation starts.
The whole flume is 18 meter length while only 1.25 m of it was in use.

*1. Why have not authors installed the slope at the very end of the flume to bringing the travel time of the reflected wave (and hence the duration of the "clean" incident wave field) to above 20 seconds?*

We did not install the slope at the end of the channel for technical reasons. The configuration of the channel bottom does not make it possible to have the desired slopes. You can see in the schema of channel below

[Figure]

*2. Can the authors comment on the influence of incident wave field irregularity on the results?*

We did not notice an influence of the irregularity of the incident wave field on the results. We checked the reproducibility of our different measurements several times.

Example: for the diagram $(a_L, f)$ fig.5 for each amplitude we did our experiments by increasing the frequency and then we repeated the experiment by decreasing the frequency. We got the same points.

***Stability threshold.***
*3. Can the discrepancy in Fig. 8 be explained by formation of the edge waves near the panel of the wave maker? Can that effect be evaluated?*

No this discrepancy in Fig. 8 cannot be explained by the formation of the edge waves near the panel of the corrugating machine. Because we did not observe any burr waves near the panel

of the wavemaker. The only affect, the amplitude of surface waves decreases when the edges waves occurs at the beach. You can see the signal from probe P1 near the panel of the wavemaker.

The theoretical values are larger than experimental ones by approximately 30%.

[Figure]

*4. It looks as the use of the whole length of the wave flume mentioned above could have helped decoupling the edge waves formed at the slope and at those formed at the wave maker.*

*It would be good to compare the rate of edge wave growth (not only the stability range) with the results predicted by (10).*

The subharmonic instability described above is investigated in the flume for different values of $(a_L, f)$, where $a_L$ is an amplitude of surface waves in the vicinity of the wavemaker. For $f$=1.08Hz and $a_L \approx a_0$, the edge waves growth exponentially and in this way we measure the growth rate $\gamma$, and we compared this rate with the results predicted by :

$$\mu = a_0 \frac{\omega^3}{4g\beta^2} S(\beta)$$

| $a_L$ [cm] | $\gamma$ [S⁻¹] | $\mu$[S⁻¹] |
|---|---|---|
| 0.179 | 0.000 | 0.023 |
| 0.277 | 0.026 | 0.036 |
| 0.366 | 0.057 | 0.048 |
| 0.469 | 0.079 | 0.061 |
| 0.572 | 0.081 | 0.075 |
| 0.663 | 0.092 | 0.087 |
| 0.740 | 0.072 | 0.097 |

We have made a display error on the ordinate axis of Figure 6a. Here is the figure after modifications

[Figure]

**Turbulence.**

5. *From the measurements, it would be good to calculate the Energy Dissipation Rate and compare it with other typical powers present in the system such as power pumped into various wave harmonics.*

We have study the energy of wave propagating in the flume. We compare wave energy near the wave maker with wave energy at shore. The wave energy (energy on a unit length in the direction transversal in the direction of wave propagation) is estimated as follows:

$$E = \frac{\rho g}{2} C_{gr} \int (\eta - <\bar{\eta}>)^2 dt$$

where $C_{gr} = \frac{d\omega}{dK}$ is the group velocity of harmonic component corresponding to the peak frequency $f$, $g$ for acceleration of gravity, $\rho$ water density, $\eta$ and $<\bar{\eta}>$ are free surface displacement and mean water level, respectively.

Typical dependences of E2 (energy at shore) on E1 (energy near the wave maker) are shown in Fig. 10 for different amplitude of excitation for $f = 1.06$Hz

We have explained in the text (page 8) this dependence of wave propagating energy

[Figure]

*Figure 10. Dependence of wave propagating energy (f =1.06 Hz) E2 (energy at shore) on E1 (energy near the wave maker) for different amplitude of excitation $a_L$*

*6. It would make a good illustration if the Authors present the power spectrum of the turbulent flow in log-log scale.*

We have modified figure and we have inserted the power spectrum of the turbulent flow in log-log scale (fig 9c)

[Figure]

*Figure 9. (c) Power spectrum of the turbulent flow in log-log scale for signal with $a_L=1$ cm, f=1.06 Hz*

***Minor points***
*P. 4, Line 18: probably, a cylindrical lens not spherical.*

We have corrected this misprint.

*Fig. 5: Do markers easier to distinguish by making, say, diamonds empty and circles filled.*

We prefer to keep this format of markers, the other format is less visible

*Fig. 6, caption: Turbulent Kinetic Energy not kinematic, right?*
We have corrected this misprint.

*P. 6, line 10: "The frequency of the zero edge wave mode Omega_0 has a minimal dissipation"*
*The reviewer does not understand relation of that to the formula (11). Why dissipation?*

We have change this sentence.

**Detailed response to reviewer 1**

*Page 1, write Douady instead of Dauady*

We have corrected this misprint.

*Page 2, line 7, change "a question" with " an open question";*

We have change this sentence.

[revised manuscript text omitted]